# Corporate Responsibility in the Digital Era

**Martin Wynn** [1,*] and **Peter Jones** [2]

1   School of Computing and Engineering, University of Gloucestershire, Cheltenham GL502RH, UK
2   The Business School, University of Gloucestershire, Cheltenham GL502RH, UK; pjones@glos.ac.uk
*   Correspondence: mwynn@glos.ac.uk

**Abstract:** As the digital era advances, many industries continue to expand their use of digital technologies to support company operations, notably at the customer interface, bringing new commercial opportunities and increased efficiencies. However, there are new sets of responsibilities associated with the deployment of these technologies, encompassed within the emerging concept of corporate digital responsibility (CDR), which to date has received little attention in the academic literature. This exploratory paper thus looks to make a small contribution to addressing this gap in the literature. The paper adopts a qualitative, inductive research method, employing an initial scoping literature review followed by two case studies. Based on the research findings, a simple model of CDR parameters is put forward. The article includes a discussion of a number of emergent issues—fair and equitable access, personal and social well-being, environmental implications, and cross-supply chain complexities—and a conclusion that summarises the main findings and suggests possible directions for future research.

**Keywords:** corporate digital responsibility; CDR; digital technologies; corporate social responsibility; CSR; customer interface; case studies





## 1. Introduction

In today's boardrooms, most senior executives would recognise four main dimensions of corporate responsibility—environmental, ethical, philanthropic and economic, usually grouped together under the umbrella term "corporate social responsibility" (CSR). In the digital era, however, as new technologies are ever more widely deployed, new sets of responsibilities are becoming increasingly evident. Bednarova and Serpeninova [1] (p. 1), for example, argued that "although digitalisation has led to a significant increase in efficiency, it raises certain concerns related to privacy, data protection and other human rights, which might be at stake when huge amounts of data are being collected and processed". The issue was highlighted by the recent call by The Future of Life Institute's open letter, signed by many business leaders, including Elon Musk, proposing a six-month precautionary pause on artificial intelligence (AI) development. "The signatories worry that AI labs are 'locked in an out-of-control race' to develop and deploy increasingly powerful systems that no one—including their creators—can understand, predict or control" [2] (p. 9). This is one—albeit much publicised—example of the need for, and value of, corporate digital responsibility (CDR), which can be defined as "the set of shared values and norms guiding an organization's operations with respect to the creation and operation of digital technology and data" [3] (p. 876).

Although few companies have publicly reported on how they are approaching their digital responsibilities, research into CDR is attracting attention in the business, management and information systems literatures [3–5]. This exploratory paper looks to build on previously published work to propose a simple model encapsulating the main parameters of CDR that emerged from case studies of Walmart and Deutsche Telekom (two major enterprises in the service sector) and to broaden the discussion to include issues relating to the policy and practice of CDR. CDR is still an emerging concept, and it may developed in

slightly different directions, and may become more complicated as it reflects variations in national and international legislation and strategic autonomy, as well as current practice on how user data companies access and store data in different jurisdictions. Within this evolving conceptual and operational environment, the paper offers some initial insights into how two major international companies are approaching the management of their digital responsibilities.

The article comprises six sections. Following this introduction, some of the relevant literature is reviewed, and then the research methodology is outlined. Section 4 contains the two case studies and is followed by an analysis and discussion of emergent issues in Section 5. The concluding section draws together some key elements of the paper and points out some possible future areas of research in this field of study.

## 2. Relevant Literature

CDR can be seen as part of the wider concept of CSR, which has been recognised and reported upon in industry for several decades. Wade [6], for example, took this view, arguing that CDR is a subset of CSR. Van der Merwe and Al Achkar [7] described CDR as one part of an overall CSR model but maintained that the focus on technology application and its repercussions warrants a clear distinction between the two concepts. Mihale-Wilson et al. [8] (p. 128) point this out: "due to the complexity that technology adds to corporate responsibility and the fact that managing the consequences and the opportunities that technologies can bring about requires a strong technological focus, it seems appropriate to view CDR as distinct from CSR".

The scope of CDR is wide-ranging and overlaps with the other dimensions of CSR noted above, having social, economic, ethical, and environmental—as well as technological components. In this context, France Strategie [9] (para. 5) argued that a digitally responsible company should respond to several major challenges, including regulatory responsibility, linked to data protection and compliance with the GDPR and sectoral regulations; ethical responsibility, linked to artificial intelligence (AI) software; societal responsibility, related to data management, the transformation of working methods, the type of data sharing and the inclusion of all; and environmental responsibility, related to the use of data in considering the environmental impacts of business activities.

Elliot et al. [10] (p. 184) concluded that CDR was "fulfilling the corporate rationalisers' role in representing community interests to inform 'good' digital corporate actions and digital sustainability via collaborative guidance on addressing social, economic, and ecological impacts on digital society". Within this context, Herden et al. [4] researched how CDR provides organisations with the opportunity to win the trust of their stakeholders, as well as to gain a competitive advantage in the marketplace. They concluded that there was not only a need but also a potential advantage for companies to implement a CDR strategy to address the threats and embrace the opportunities presented by digital technologies. They argued, nevertheless, that "as each company has unique goals, business strategies and CDR needs, an individual CDR strategy is essential" [4] (p. 25) and that companies need to regularly revisit and revise their CDR policies and organisational structures, given the continual evolution of digital technologies.

Lobschat et al. [3] focused on the development of CDR compliant behaviour in company operations. The authors found that "for a business to be digitally responsible, its managers and employees must align their behaviours with specific norms established by the organization to achieve CDR" (p. 875). They concluded that this may lead to cross-company disruption, entailing organisational restructuring, new staff competencies and re-training, and the redesign of data management and communication processes. The research of Jelovac et al. [11] similarly examined the impact of CDR on building digital trust and responsible corporate governance in companies. They concluded that "the best response to building and maintaining trust is, in our opinion, the building of a new modern business and organizational CDR culture" (p. 494).

In the context of service industries, Wirtz et al. [12] maintained that the concept of CDR had received little attention, but pointed out its criticality given the access that service sector companies have to vast streams of customer data. The authors argued that service companies should look to ensure that CDR issues were addressed, particularly in their supply chains, with their business partners, and where secondary users had access to their customer data. In a similar vein, but in the context of online banking, Liyanaarachchi et al. [13] concluded that online operators should look to incorporate CDR into their operations to help reduce consumers' exposure to data privacy contravention. They suggested that CDR should be a central component of organisational strategy, and that banks should limit their exposure to data breaches, which could damage brand equity. Wirtz et al. [12] proposed a number of related research agendas including: how do a company's CDR behaviours and practices influence engagement, trust and loyalty; which governance procedures are most effective in encouraging CDR compliance amongst a company's business partners; how technological developments, for example, in AI, encryption, and blockchain, can be employed to enhance a company's CDR performance; and, more generally, on the costs and benefits of CDR.

Mihale-Wilson et al. [8] (p. 130) highlighted two main complementary research directions that could profitably be pursued: "the conceptualisation and operationalisation of CDR" and an analysis of the "suitability and effectiveness of different CDR measures". As regards the second of these research initiatives, the authors recognised the need to investigate the suitability and the effectiveness of a range of CDR activities for different stakeholder groups across a variety of types of business. The authors emphasised that a company's ability to successfully introduce CDR might be determined, in part at least, by the industry in which that company operates and that the measures through which CDR could be introduced in highly digitised industries, for example, might differ significantly from those in less digitised ones. They argued that the scholarly conceptualisation of CDR was still in its infancy and that their work sought to contribute to CDR theory by providing a more in-depth assessment and understanding of the concept. They theorised the link between the proposed CDR norms and digitisation challenges and argued these norms could serve as a preliminary conceptual framework for CDR. "Access" concerns consumers having access to basic digital goods and services; "information and transparency" refers to consumers having appropriate information availability so they can be informed according to their individual wishes and needs; "economic interests" are described as the protection and promotion of the consumers' economic interests; while "privacy and data security" concerns the protection of consumers' privacy and the free flow of information, as well as the offer of protected and secure payment mechanisms.

In a similar vein, Isik and Wade [14] identified four components of digital corporate responsibility, these being social, economic, technical, and environmental components. Social CDR, for example, is seen to include "ensuring data protection for employees, customers and other stakeholders" (para. 2), while economic CDR includes "using technology responsibly to replace jobs, done by people" (para. 3) and sharing the economic benefits of digitalization with society through things such as taxation. Technical CDR involves ensuring that the production of digital technologies does not harm society, and environmental CDR looks to extend the life span of technology and encourage responsible power consumption practices.

Within this context, this article addresses two research questions:

RQ1. What are the main parameters of CDR that are evidenced in the two industry case studies?

RQ2. What further issues emerge as regards the operation of CDR policies and practice in the industry case studies?

## 3. Research Method

The paper employs a case study approach to illustrate some of the ways in which two major companies—Walmart and Deutsche Telekom—are publicly addressing CDR.

These two major companies in the service sector both have access to massive amounts of client and customer data and have recently posted some details of their approach to CDR on their websites. As such, this paper might best be seen as an opportunistic endeavour designed to shed some preliminary light on an issue that has received very little attention in the academic literature. Deutsche Telekom, originally established in 1995, is a German telecommunications company, and it is the largest telecommunications provider in Europe. The company has substantial shares in telecommunications companies in a number of countries, including Austria, Greece, Slovakia, Hungary, Poland, the Czech Republic, Croatia, North Macedonia, Romania, Montenegro, and the US. Walmart, originally established in 1962, is a multi-national retailer based in the US. As the world's largest retailer, Walmart operates over 10,000 outlets and has over 2 million employees. The company trades from a range of formats, including superstores, discount stores and convenience stores, and it has operations in 24 countries, including Canada, Mexico, China, as well as the US.

Prior to focusing on the case study companies, recently published academic literature and information obtained from various web sources was reviewed to provide the material presented in the Literature Review above. This was a scoping review aimed at identifying key themes that provided the basis for developing the two research questions to be addressed in the case studies. Scoping reviews "are best employed when there is limited literature to inform the research question of interest" [15] (p. 5) and can help to lay the foundations for subsequent research endeavours. The case studies were based on qualitative data drawn from material posted by Walmart and Deutsche Telekom on their corporate websites. Rowley [16] (p. 16) argued that "case studies have often been viewed as a useful tool for the preliminary, exploratory stage of a research project", and while these case studies do not offer a complete picture of how the two companies have approached CDR, the authors believe they provide some valuable insights into how CDR operates in international companies.

In developing the two case studies, the authors looked to capture the companies' approach to CDR in their own words, on a number of occasions, in the belief that such quotations help to convey corporate authenticity. Document analysis was thus the main technique used in the case studies. Bowen [17] defined this as a "procedure for reviewing or evaluating documents—both printed and electronic (computer-based and Internet-transmitted) material", noting that "like other analytical methods in qualitative research, document analysis requires that data be examined and interpreted in order to elicit meaning, gain understanding, and develop empirical knowledge" (p. 27). This helped the authors identify six emergent themes that are discussed below, this being an iterative, cyclical process involving the working and re-working of common themes and related issues. As Walsham [18] has observed "it is desirable in interpretive studies to preserve a considerable degree of openness ... ... this results in an iterative process of data collection and analysis, with initial theories being expanded, revised, or abandoned altogether" (p. 76).

## 4. Case Study Findings

The two case studies provide somewhat different perspectives on the ways in which Walmart and Deutsche Telekom are addressing CDR, but they focus on the limited information the companies currently communicate about their approaches to CDR within the public realm. The Walmart case study largely addresses policies, while the Deutsche Telecom focuses more on values and supporting initiatives. Together, however, the case studies provide new insights into the various ways in which the two major companies claim to be addressing CDR.

### 4.1. Walmart

In addressing "digital citizenship", namely, the "ethical use of data and responsible use of technology", Walmart [19] (para. 1) claimed "we seek to build and maintain the trust of customers, associates and communities with respect to our use of technology and data,

in line with our values of service, excellence, integrity, and respect for the individual". The company suggested that "almost every aspect of Walmart's business relies on the use of technology and data, including business sensitive and proprietary data as well as personal data from our customers", and that "our customers trust us to use their data to help provide them with relevant and exciting products, services, shopping experiences and innovative ways to help them save money and live better" [19] (para. 3). Furthermore, the company emphasised its belief that "our commitment to ethical use of data and technology helps build customer trust in our brand and products and helps mitigate the risks of improper data and technology practices" [19] (para. 3).

More specifically, Walmart [19] (para. 5) claimed that its "digital trust commitments provide a foundation for the company to earn and maintain customer trust in an omni-channel, data- and technology-driven world", and that these commitments were built upon the company's core values, namely, "service, excellence, integrity", and "respect". The aim is to put these commitments into practice in four key areas, namely "promoting fairness", "protecting privacy", "data, records and information management", and "cybersecurity and information security" [19] (para. 6).

In addressing promoting fairness, Walmart [19] (para. 7) claimed that its "Digital Citizenship Team helps the company to achieve our digital trust commitments as the company develops and implements new technologies, new services and new ways to capture and use data". By way of two simple illustrations of its work in this area, the company outlined its development of a framework to evaluate AI and machine learning, and its work in operationalising its digital trust commitments. In evaluating AI and machine learning, the focus is on mitigating bias and promoting fairness in the development and use of these tools, while in operationalising digital trust, the company's aspirations for the new digital technologies are that they should be "flexible and scalable", their usage should be "clear and accessible", and that they "should be designed, evaluated and tested to reduce bias, both implicit and actual" and to "increase transparency".

In looking to promote privacy, the Audit Committee of Walmart's Board of Directors, "oversees risks related to data privacy as part of its information and security and cybersecurity oversight" and the company's digital citizenship team "helps to oversee Walmart's compliance with our privacy policies and applicable laws" [19] (para. 10). At the same time, Walmart [19] (para. 11) claimed "we aim to provide customers, associates and other stakeholders with clear, prominent and easily accessible information on how we collect, use, share and protect personal information". More generally, the company "tracks emerging data privacy laws and implements compliance programs across the global enterprise", and claimed to have "dedicated professionals that focus on compliance with laws enabling our customers to request information under various data subject access request laws that exist today and that may be passed in the coming years", and to "have designed our processes and systems to be as resilient as they can be to accommodate different coming state laws and meet the expectations of our customers and regulators about data transparency" [19] (para. 14).

The company has clear governance structures for cybersecurity and information security and reported that its "Information Security Management Policy" is the foundation of its information security programme, and that "this policy applies everywhere Walmart data is stored or processed—within Walmart and outside it—and speaks to the security requirements for assessments, account and device security; personnel security; and awareness and training" [19] (para. 27). Additional policies include "escalation processes that associates can follow should they notice something suspicious", and here "associates are required to report known or suspected violations of the policies" [19] (para. 27). At the same time, Walmart [19] (para. 28) reported that "vendors that have access to Walmart information are required to manage such information in accordance with laws and appropriate privacy and security standards", and that "standards are applied on a per contact basis and include requirements to report to Walmart any incidents in which Walmart information systems are compromised".

*4.2. Deutsche Telekom*

In his introduction to "Corporate Digital Responsibility @Deutsche Telekom", Timotheus Hottges, Chair of the Board of Management, argued that "responsible digitization represents the extension of our practiced corporate responsibility into an increasingly digitalized world. Based on this conviction, we design our internal processes, business activities and business relationships; we adapt our product portfolio and service offerings; we stand up for community, and campaign against the division of society" [20] (p. 4). More specifically, the company recognise that "digital trends are affecting and changing all our business processes" and argued that "we consider digital responsibility to be the conscious decision to pursue ethically sustainable and responsible actions within the digital transformation" [20] (p. 7) and that this approach was part of the company's culture and the values it embraced.

In seeing "digitalization as opportunity", as it means "more people can participate in public discourse" with "hardly any limits to communication and understanding" [20] (p. 8), the company emphasised its awareness "that we have to face the corresponding risks". The company recognises that conflicting values and dilemmas needed to be addressed in society and that it needed to make digitalisation compatible with its values and to find the optimal solutions to shape responsible digital transformation. In addressing these conflicting values, Deutsche Telekom also recognised that it could not shape this retransformation on its own, but that it was a task for society as a whole, which must involve a wide range of stakeholders.

The company argued that its approach to digital responsibility was focussed on "human-centred technology" [20] (p. 11) and built on a series of foundations, namely, laws and regulations, human rights, and culture and values, and two principles, namely data privacy and security and transparency and dialogue. In addressing laws and regulations, Deutsche Telekom emphasised that the company not only complied with minimum legal standards, that it assessed, and externally reported on, but that it also contributed a variety of initiatives focused on digital ethics as part of its perceived role as a dialogue partner in the digital world. Deutsche Telekom emphasised is commitment to United Nations Guiding Principles on Business and Human Rights, and that the company stood for digital sovereignty, freedom from discrimination and freedom of expression for its employees and customers. Here, the company recognised that technology has become deeply entwined in all aspects of life, and that as such, it influences economic and social activity in a wider context. In addressing culture and values, Deutsche Telekom claimed that, as a global company, not only did it recognise cultural differences, but that it looked to leverage them to achieve success.

Data privacy and security is the first of Deutsche Telekom's two principles of digital responsibility, and here the company emphasises that it stood for security and the responsible handling of data, not least in that it argued that its customers, employees, shareholders, the regulatory authorities, and the general public rightly expected it to handle the data they entrust to it with care. At the same time, Deutsche Telekom argued that, in addition to data privacy and protection, transparency in how data were used and processed was a central issue, and that the company reported on the data it used, and for what purposes, how long it is retained, and under what special circumstances any disclosures are made. In pursuing the second principle, namely transparency and dialogue, Deutsche Telekom argued that it looked to shape the dialogue about the opportunities and risks of digitalisation, and to this end, that it communicated with its customers and maintained an ongoing dialogue with its employees, and that all its communications were characterised by respect, integrity and transparency.

In focusing on "technologies for people and with people" Deutsche Telekom [20] (p. 11) outlined its four action areas related to digital responsibility, namely digital ethics, digital participation, future work, and climate and resource protection. The company claimed to be a pioneer in digital ethics and more specifically, reported on its support for project managers, data scientists, and programmers as part of its ethics by design approach

during AI development, and on its support for the design of trustworthy products for customers. In emphasising its commitment to digital participation, Deutsche Telekom [20] (p. 23) claimed "we want everyone to take part in the digital society", recognising that "social participation in the digital sphere requires, access, affordability and skills", and that "people must be motivated to take part and live together in the digital world according to democratic rules".

In addressing future work, Deutsche Telekom emphasised that the workplace was changing rapidly, not least in that employees' willingness to learn and change were fast becoming core competencies. At the same time employees were increasingly expecting their employers to offer them more personal freedom, greater flexibility and less limitation to specific workplace locations. Both of these forces are in evidence as competent, committed and entrepreneurial oriented employees are taking greater responsibility for their work. As regards the environment impacts of digitalisation, Deutsche Telekom emphasised its commitment to climate protection and resource conservation, and here the company claims that its principal focus is on reducing its own environmental impact and that of its customers.

## 5. Analysis and Discussion

This section draws on the case study findings and is structured around the two main research questions. In Section 5.1, a simple model representing CDR is presented, building upon issues evidenced in the case studies and supporting literature. In Section 5.2, six main themes are identified and discussed that make reference to the model, material from the case studies and relevant literature.

### 5.1. RQ1. What Are the Main Parameters of CDR That Are Evidenced in the Industry Case Studies?

Many major enterprises have looked to access a seemingly ever wider range of business opportunities by harnessing digital technologies to transform all areas of their operations, but the vast majority of them have been much slower to publicly acknowledge, and address, the new set of responsibilities associated with the introduction of these technologies. This makes the case studies of Walmart and Deutsche Telekom of particular interest, because they confirm and build upon the vast majority of issues evidenced in the literature review and support the development of a framework comprising parameters of CDR as a subset within the broader CSR agenda (Figure 1).

There are a range of actors and stakeholders involved in CDR, but employees and customers are arguably those most affected. Whilst a number of stakeholders interact with most of these parameters, it is company employees who are at the forefront of CDR related upskilling and redeployment, cultural change and process re-design, and measures and policies aimed at greater data transparency and access. Customers are central to data protection, privacy and security issues, and to the end-to-end theme of building trust across the organisation's interface with the customer through a growing range of digital engagement technologies (social media, chatbots, analytics, big data, AI). Broader issues span the divide between CDR and CSR, notably the environmental impacts of digital technologies, ethical issues associated with technology deployment, and the need for business alignment with new norms and regulations regarding some of the parameters discussed above. This simple model can be set alongside those developed by Lobschat et al. [3] and Wade [6], both of which have been reviewed by Jones and Comfort [21].

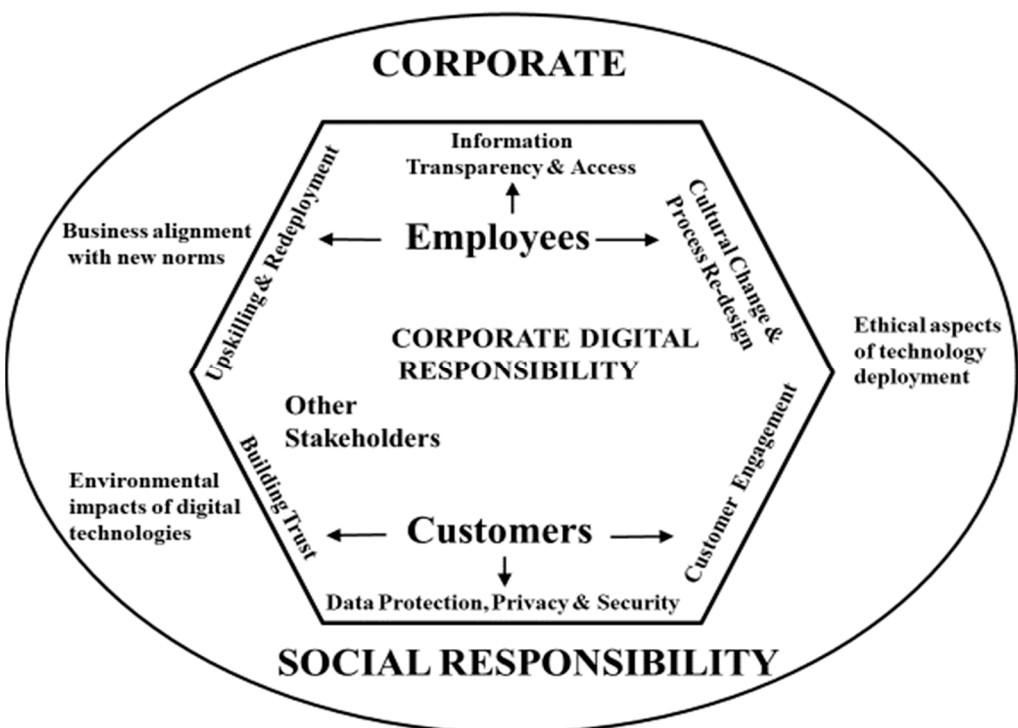

**Figure 1.** Main parameters of corporate digital responsibility.

Whilst few other major companies have fully embraced the concept of CDR, many have recognised and acted upon some of these parameters, and report on them in their CSR or Environmental, Social and Governance (ESG) reports. For example, the Dutch multinational Ahold Delhaize [22] includes a section on data privacy in its ESG report, in which it sets out five principles "that guide how Ahold Delhaize and its brands manage personal data" (para. 2), noting "customers, associates and business partners entrust our businesses with their personal data, and we must safeguard this information at all times" (para. 1). Tesco [23] have created a "privacy centre", claiming "we take the responsibility that comes with being entrusted with your personal data very seriously, and we're committed to respecting your rights regarding the use and security of your personal data" and that the company wished to provide "clear and transparent information about how we collect, use and protect your personal data, the circumstances where we may share your personal data and your rights in relation to your personal data" (para. 3). There are many other examples of how strands of CDR policy are evidenced in company ESG and CSR reports, but as yet few are piecing them together in an integrated CDR strategy.

*5.2. RQ2. What Further Issues Emerge as Regards the Operation of CDR Policies and Practice in the Industry Case Studies?*

In addition to the above noted model, these cases highlight six interlinked sets of issues that provide insights beyond this framework.

Firstly, there is a set of issues in and around corporate commitments to providing fair and equitable access to digital technologies and in enabling access to digital society. Here, the digital divide, simply defined as disparities in access to, and use of, digital technologies, can be an important issue. While both Deutsche Telekom and Walmart operate largely, although not entirely, within developed economies, where digital access is generally good, such access is not universal within such economies and this in turn can exclude some individuals from the flexible shopping and purchasing powers offered, for example, by Walmart, and the wide range of communication and commercial opportunities and social media facilities, offered by Deutsche Telekom. Corporate commitments to improving digital access, largely in developed economies will only serve to exacerbate inequalities between

those sections of society who benefit from the seemingly ever wider range of services such increased access brings and those, particularly in the less developed world, who continue to have lower levels of access to digital technologies.

Secondly, there are issue about digital technologies and personal and social wellbeing. While both Deutsche Telekom and Walmart are keen to emphasise the widespread social benefits which digital technologies offer, concerns have been widely expressed that the overuse of, and increasing dependence on, digital-technology-enabled devices can have a negative impact on physical and mental wellbeing. On the physical side, the extended use of smartphones, computer screen and tablets, can cause eye strain and lead to blurred vision and head and neck pain, may also cause poor posture, and can reduce sleep quality. Psychological impacts can include addiction, depression and anxiety. More substantively Burr et al. [24] (p. 2313) argued that "the rapid deployment of digital technologies and their uptake by society has modified our relationships to ourselves, each other, and our environment. As a result, our individual and social well-being is now intimately connected with the state of our information environment and the digital technologies that mediate our interaction with it, which poses pressing ethical questions concerning the impact of digital technologies on our wellbeing".

Thirdly, the increasing use of digital technologies has important environmental dimensions. One body of opinion suggests harnessing digital technology will have a vital role to play in the transition to a sustainable future. The United Nations Environment Programme [25] (para. 5), for example, have suggested that "a digital ecosystem of data platforms will be crucial to helping the world understand and combat a host of environmental hazards, from air pollution to methane emissions". However, the increasing adoption of digital technologies might be seen to be the antithesis of sustainability, and more specifically, of sustainable consumption. DataCamp [26] (para. 2), for example, pointed out that "an increasing number of studies are alerting us to the significant climate and environmental impact of our digital activities". The data centres, for example, which drive the digital technologies, and on which they ultimately depend, are major energy users and as such contribute to greenhouse gas emissions and to climate change. In addressing these environmental impacts, DataCamp [26] (para. 3) argued "from carbon-intensive mining activities and manufacturing operations to increasing electricity demand from data centres and product obsolescence that results in tonnes of e-waste, a comprehensive environmental audit of digital technologies is crucial to understanding the sector's impact on climate change and biodiversity loss". Neither Deutsche Telekom nor Walmart make any mention of such environmental audits in their public disclosures of their approaches to CDR.

Fourthly, there are thorny issues about whose interests are best served by CDR. Lobschat et al. [3] (p. 879), for example, argued "the multisided natures of many markets for digital products and services makes the assessment of beneficence for all involved stakeholders complex". On the one hand companies, such as Deutsche Telekom and Walmart, in the service sector, have capitalised on their deployment of digital technologies to offer a seemingly ever wider range of services and facilities to their customers, and by and large, their customers have enthusiastically adopted these services and facilities. On the other hand, Van der Merwe and Al Achkar [7] (para. 25) present an alternative perspective, suggesting that CDR "offers corporations an opportunity to build a cover for unethical behaviours and practices".

Fifthly, a number of potential internal contradictions can be identified within the companies' approach to and operation of CDR policies, notably as regards customer data. Whilst service industry companies want to use their customer data in a variety of ways to drive their businesses, their customers are looking to protect their privacy and their rights, as well as harness the benefits and conveniences many digital technologies offer. This is evidenced by trade-offs between the promotion and management of the economic benefits of digital technologies and the companies' responsibilities and relationships with customers, individuals, and other stakeholders. Consumers are unlikely to ever willingly relinquish the flexible purchasing powers offered by digital technologies, which further

complicates company commitments to CDR. Another aspect here is that employees may face the possibility that digital technologies will be deployed in a wider range of contexts, particularly at the customer interface, leading to a reduction or replacement of jobs at store level and in logistics operations, and even in head offices.

Sixthly, there are issues relating to the operation of CDR policies internationally and across supply chains. While large companies such as Walmart and Deutsche Telekom can establish corporate policies on CDR, there may be different interpretations of those policies in different countries, which may, in turn, reflect not only the lack of clarity concerning the precise meaning of CDR but also the different cultural environments and political jurisdictions, and the different regulatory environments within those jurisdictions, in which international companies operate. The relationships between these companies and their vast number of suppliers add further complexity in that, in addition to the global nature of supply chains, the controls on digital technologies and data are at least one step removed from direct corporate control. One significant aspect here is cybersecurity, particularly with technology product suppliers such as Deutsche Telekom, where the true origin of component parts is not always evident, hampering a realistic assessment of the cyber security risk of such products [27]. Independent auditing of CDR policies and activities may be an option for large enterprises to publicly confirm and legitimise their corporate CDR policies within their international operations and supply chains. However, some critics have suggested that the audit process is flawed and, at worst, that it may be self-serving. LeBaron et al. [28] (p. 958), for example, argued that "the growing adoption of auditing as a governance tool is a puzzling trend, given two decades of evidence that audit programs generally fail to detect or correct labor and environmental problems in global supply chains".

## 6. Conclusions

This paper has provided some insights into how Walmart and Deutsche Telekom are accommodating CDR within their overall business strategies, and along with a review of the relevant literature, has provided a simple model of the main parameters of CDR, which practitioners may find useful in reviewing the case for, and practicalities of, developing their own CDR norms and policies, and possibly coordinating them within broader CSR strategies. More specifically, the paper has highlighted and discussed a wide range of themes that are informing what is a generally positive approach to CDR currently being adopted by Walmart and Deutsch Telekom.

These themes include data privacy and security, information access, customer engagement and the building of trust, and aspects relating to the personal and social well-being of employees. In the wider context of CSR, there are issues around the ethical implications of digital technology deployment, as evidenced in the current debate on the future of AI, and the environmental impacts of digital technologies. While Deutsche Telekom look to include some environmental responsibilities as part of their approach to CDR, Walmart do not, and perhaps not surprisingly, neither company address the thorny issue about whose interests are best served by CDR. Many large companies address environmental and climate change concerns in their CSR report, and they may well gradually look to add such concerns to their commitment to CDR, but as companies increasingly embrace digital technologies, a balanced treatment of whose interests are best served by CDR seems likely to remain outside any future CDR reporting processes. Here, the suggestion by Van der Merwe and Al Achkar [7] (para. 25) that CDR could be used for "whitewashing and regulatory capture" must be a continuing cause of concern for many stakeholders.

This paper has a number of limitations. It is, in the main, based on two case studies of large international companies, using material drawn from Internet sources. Wider generalisations should thus be treated with caution. Rather, as Flyvbjerg [29] noted, each case should focus on investigating and assessing the dynamics of the case itself, producing "concrete, context-dependent knowledge" (p. 223). At the same time, the two case studies report on what is effectively the initial development of CDR without detailed reference to

what are still very much national/international regulatory policies or to how CDR may also be shaped by the need for companies to be competitive within national/international business environments. A further limitation is that while the case studies offer illustrations of how the two companies are currently approaching CDR, they do not provide a comprehensive picture, or a detailed analysis, of the development and workings of current CDR policies.

Such, albeit important, limitations aside, this exploratory paper may provide a platform for future research agendas. There remains much work to be conducted, for example, in developing an appropriate theoretical framework to underpin CDR research and its implementation in practice. This could build upon the simple model included here, and others noted in the extant literature. This might include consideration of how CDR fits within the broader CSR concept and if digital responsibility should now be formally recognised as a fifth dimension of CSR. In addition, empirical studies based on primary data sources could explore how companies are addressing CDR within and across their supply chains. This mirrors the call for new research within sustainability studies in general to examine cross-supply chain issues, notably for the transitioning to circular economy practices [30]. Other studies could investigate how, if at all, customers have been involved in the CDR development process, their levels of trust in the company statements about privacy and the security of financial and personal data, and the extent to which such levels of trust influence patronage.

**Author Contributions:** Conceptualization, M.W. and P.J.; methodology, M.W. and P.J.; formal analysis, M.W. and P.J.; investigation, M.W. and P.J.; writing—original draft preparation, M.W. and P.J.; writing—review and editing, M.W. and P.J.; visualization, M.W. and P.J. All authors have read and agreed to the published version of the manuscript.

**Funding:** This research received no external funding.

**Data Availability Statement:** Data used in the case studies is available in the sources cited in the text.

**Conflicts of Interest:** The authors declare no conflict of interest.

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
