# Peer review of "Corporate Responsibility in the Digital Era"

_information, doi:10.3390/info14060324_

Round 1

Reviewer 1 Report

Thank you for the opportunity to review the paper. I find the paper well structured, topical, and it presents a clear evidence-based conclusions. As the paper takes a focal firm perspective to its topics, I would like to raise a few perspectives and comments for consideration for the authors. First. currently we can argue that there are three major approaches to data in global businesses: the US model where the firms own and control data collected from users; the Chinese government-control approach; and the European model that emphasizes consumer/user rights (see e.g. Feijoo et al (2020), Telecommunications Policy). These approaches are difficult to combine into single unified CDR approach. Second, at the same level of discussion, strategic autonomy, sovereignty, and maintaining democracy are issues that are increasingly important issues at the national level when it comes to digital technologies, especially critical infrastructures like telecoms or other platforms. This brings about themes such as privacy, security, and resilience of digital systems. Third (and related to points one and two), the regulatory domains for digital services is getting more complicated, especially within EU, for example, related to Deutche Telekom, there are lots of new emerging regulations influencing the way how CDR is conducted or should be approached. Fourth, as a consequence of technology convergence, the the business/digital ecosystems in multisided platforms/markets will have direct effects on competition. The emergence of walled gardens and winner-takes-it-all consequences of platforms may require ecosystem-level considerations when developing CDR policies. I would like the authors to consider how to bring these themes into the article.

Reviewer 2 Report

The manuscript title is quite general. Too many quotes for easy-to-paraphrase text portions. Simply displaying other authors’ ideas is not enough: a thorough analysis is needed. Try and replace the not-peer reviewed sources. ‘Some authors have focused on how CDR impacts people and processes within the organisation, whilst others have looked more broadly at the customer interface and how
CDR encompasses customer perceptions and engagement.’ – but you mention a single source. Some sentences are poorly constructed. E.g., ‘The culture of the organisation was researched by Lobschat et al. [3], focusing on the development of CDR compliant behaviour in company operations.’ – as if Lobschat et al. were the only authors who researched the culture of the organization. Also, the culture of the organization did not focus on the development…, but Lobschat et al. ‘recently published academic literature and information obtained from various web sources was reviewed to provide the material presented in the Literature Review above.’ – a rigorous methodology is needed as regards article selection. The discussions require more structure and there is a need of offering a clear assessment of reviewed literature.

Some sentences are poorly constructed. E.g., ‘The culture of the organisation was researched by Lobschat et al. [3], focusing on the development of CDR compliant behaviour in company operations.’ – as if Lobschat et al. were the only authors who researched the culture of the organization. Also, the culture of the organization did not focus on the development…, but Lobschat et al.

Reviewer 3 Report

This seminal paper deals with a brand new topic connected with CSR that has been barely studied in the literature. Both the literature used and the methodology, using a qualitative, inductive research method by analyzing Walmart and Deutsche Telekom’s CDR strategies, make the analysis relatively limited, so the authors make a small contribution to addressing the CDR gap in the literature. Still, I miss the conclusions of their study, as in the conclusions section, the authors include the limitations, the applications for the industry, and the future lines of research, but the conclusions linked to the research questions are missing.

Round 2

Reviewer 2 Report

I'm afraid this research does not contribute to the literature significantly. Most comments have been ignored.